# Influence of Sintering on Thermal, Mechanical and Technological Properties of Glass Foams Produced from Agro-Industrial Residues

**DOI:** 10.3390/ma15196669

**Published:** 2022-09-26

**Authors:** Fernando Antonio da Silva Fernandes, Dayriane do Socorro de Oliveira Costa, João Adriano Rossignolo

**Affiliations:** 1Department of Biosystems Engineering, University of São Paulo, USP, Av. Duque de Caxias Norte, 225, Pirassununga 13635-900, SP, Brazil; 2Department of Engineering, Federal University of Pará—Campus Salinópolis, Rua Raimundo Santana Cruz, S/N, Bairro São Tomé, Salinópolis 68721-000, PA, Brazil; 3Department of Engineering, Federal University of Rio de Janeiro, Salinópolis 68721-000, PA, Brazil

**Keywords:** agro-industrial waste, glass foam, thermal insulation, circular economy

## Abstract

This study investigates the technological, thermal, mechanical, and technological properties of glass foams produced with soda-lime glass residues and rice husk ash sintered at 850–950 °C. The results for apparent density (0.28–0.30 g/cm^3^), porosity (82–87 ± 4%), compressive strength (1.18 ± 0.03–1.25 ± 0.03 MPa), and thermal conductivity (0.283–0.326 W/mK) are within the limits for commercial foams. The volumetric expansion potential and low thermal conductivity of the glass foams produced favor their use as thermal insulating materials in coat walls, thus improving thermal comfort in the construction sector. The results of X-ray fluorescence show that the foam glass is of the soda-lime type (SiO_2_, Na_2_O, and CaO), the rice husk ash is rich in SiO_2_, CaO, Na_2_O, Al_2_O_3_, K_2_O and Fe_2_O_3_, and the calcium carbonate is rich in CaO. The glass foams produced in this study are promising because they present more economical and efficient manufacturing, resulting in lightweight materials with thermal insulating properties that can be used in the construction sector. These glass foams also reduce the consumption of natural and synthetic raw materials, adding value to the waste used in this study by transforming them into co-products, thus favoring the economic circulation of the region.

## 1. Introduction

Today, there is a growing worldwide concern regarding the uncontrolled disposal of wastes in the environment, which makes recycling fundamental in the production of new materials [1] because it presents advantages such as decreased energy consumption and mitigation of greenhouse gas (CO_2_, SO_x_ and NO_x_) emissions [2,3,4]. Recycling can reduce the use of natural and synthetic raw materials, adding value to waste by transforming it into a co-product, and thus strengthening the circular economy of regions through its commercialization and processing [5,6,7,8].

The civil construction sector has always been involved in discussions on sustainability because of its strong environmental impact on the use and production of materials [9]. For example, temperature indices have increased in several countries because of climate change, leading to a growing demand for materials with thermal insulation properties [10,11]. In this context, glass foams have become of great interest to civil construction [12], as they are chemically inert and non-toxic, present low density, high compression resistance, better thermal insulation [13], and greater resistance to weather compared to polymeric foams, which can cause serious problems related to fire risk, short service life, and toxicity [14,15,16,17,18]. Temperature is known to have a strong influence on the production of glass foams [11,19]. They can be produced at 800–950 °C [20,21] with a low heating rate, which can influence the formation of pores [22]. Therefore, glass foams are widely used in civil construction as blocks in the insulation of ceilings and walls for both high and low temperatures, as sub-grade improvement material, or as light aggregates in concrete [12,17,23]. Several materials are suitable for manufacturing glass foams, alumina, mullite, silicon carbide, zirconia, and hydroxyapatite. Soda-lime glass accounts for 90% of all glass produced worldwide, and studies have demonstrated the advantages of its application [16], which can favor the expansion of glass foam (<450%) with the addition of CaCO_3_ as a foaming agent [13,24,25]. The soda-lime glass can represent 97% of the glass foam mass [11,26,27]. Glass foams can be designed according to their possible application by modifying the parameters of their fabrication processes, such as formation technique, amount of gas-forming agent added, etc. [28]. Therefore, from an ecological point of view, the use of glass foam is a good alternative for the civil construction sector [13,24]. The possibility of incorporating agro-industrial wastes into the manufacturing process of glass foams plays an important role in the production of this material, as it usually requires less energy to be produced than conventional materials [12]. For example, agro-industrial wastes can be used in the glass matrix as a component rich in SiO_2_, without jeopardizing the matrix formation or its properties. Among these wastes, rice husk, which corresponds to 20–22% to the whole rice mass [29], has gained importance because of its properties. In addition, rice is the most commonly consumed food in the world, and the agricultural product that generates the largest amount of residues (husks), with an estimated production of 450 million tons for 2020 [11]. Rice husk ash (RHA) originates from the process of energy generation from the sintering of rice husks [30] and corresponds to 18–20% of the whole grain weight [11]. RHA is chemically inert [31], has a high SiO_2_ concentration (>92%) [32], and has great potential to be used in the production of thermal insulating materials [33]. Moreover, depending on the sintering temperature, it may present an amorphous or crystalline structure [30,34].

This study focused on the manufacturing of more economical and efficient glass foams [19], using recycled soda-lime glass and RHA residues from the central region of Brazil as raw materials, favoring consecutive circulation and a clean and circular economy in the region.

## 2. Materials and Methods

### 2.1. Material Preparation

Initially, the glass bottles were washed with water and left to dry at room temperature. After that, they were manually crushed into <10 mm–thin pieces using a hammer and then ground in a ball mill (TS RUBENS) at 100 rpm to particle size D_90_ = 73.51 µm. RHA with D_90_ = 73.92 µm particle size resulting from the sintering of rice husks for energy generation in a red ceramic plant located in the municipality of Cristalândia (10°36′14.7″ S; 49°11′56.0″ W), state of Tocantins, Brazil, was added to the glass powder. Industrial CaCO_3_ (D_90_ = 2.76 µm) was purchased from Dinâmica/Brasil enterprise and used as foaming agent. The chemical composition analysis of the raw material used in this study was obtained by X-ray fluorescence (XRF, 1800, Shimadzu, Kyoto, Japan) and is presented in Table 1.

### 2.2. Sample Preparation

Characterization and investigation of the technological properties of the produced glass foams were performed on formulated samples: soda-lime glass (78% wt), rice husk ash (16% wt), and CaCO_3_ (6% wt). The motivation for using this formulation was to produce porous glass foams (87–93%) [11]. A total of 10 samples were investigated at each temperature to increase the reliability of the results. The raw materials were weighed on an analytical digital scale. All raw materials were weighed on a digital scale, manually homogenized for 2 min in a porcelain mortar, and granulated with the addition of water (5%) and PVA (solution containing 5% active material) (polyvinyl alcohol P.S, Dinâmica Brasil) [18] for 2 min to ensure homogeneity [11]. The resulting mixture was then placed in a stainless steel matrix (60 × 20 × 20 mm^3^) for uniaxial pressing using a hydraulic press with a 20 MPa load [35]. After pressing, the green samples were removed from the mold and allowed to dry at room temperature for 4 h. After drying, the samples were sintered in an electric muffle (EDG, 1800, Brazil) (850–900–950 °C) with a heating rate of 100 °C/min for 30 min threshold, defined in a pre-test. After sintering, the samples were cooled inside the muffle to room temperature to avoid cracking due to the thermal stress accumulated in the material’s cellular structure [17]. A sample sintered at each temperature was selected, crushed, and subjected to X-ray diffraction (XRD, Shimadzu, Japan).

### 2.3. Sample Characterization

The potential of the glass foams was evaluated through apparent density and porosity, compressive strength, microstructure evaluation, and thermal conductivity tests.
(1)$$\varepsilon \left(\%\right)=\left[1-\frac{\rho_{a}}{\rho_{t}}\right]$$

Porosity (*ε*) of the glass foams was obtained through Equation (1), where (*ρ_a_*) is apparent density and (*ρ**_t_* _=_ 2.5 g/cm^3^) is the theoretical density of the powder.

The mass and geometric dimensions of the samples were used to calculate the apparent density according to Francis and Abdel Rahman [36]. They were then cooled by inertia and characterized by pellet size (ABNT–NBR–7211) [11]. A heating rate of 10 °C/min was applied for the differential thermal (DTA; SDT–Q6000, TA Instruments) and thermogravimetric (TGA; TGA–50, Shimadzu) analyses, which were performed under a synthetic air atmosphere. The average particle size distribution of the raw materials was obtained through laser scattering (Cilas 1180). A universal testing machine (EMIC, DL–3000) was used to obtain the compressive strength values. Compressive strength was investigated in the samples (30 × 30 × 30 mm^3^) according to the ASTM C133–97R15, 2015 standard with three replicates. Samples with a dimension of 30 × 30 × 30 mm^3^ were prepared and used to determine thermal conductivity, which was obtained according to the ASTM D5334–14 standard using a surface probe (25 × 25 mm^2^) at 10^−1^ °C with a heat transfer analyzer (ISOMET Model 2114, Applied Precision, Slovakia). The chemical composition was obtained using XRF (Shimadzu XRF1800). Pore microstructure was visualized using an optical microscope (Olympus, 3Z61). The pore structure was also visually assessed between the intersections of the pore walls [37]. Volumetric expansion was monitored through recording with a portable camera (SAMSUNG Full HD 27, Samsung, China), which could store the information on a memory card and project the image in real time on a 55″ television, from which the volumetric expansion images were selected during sintering (800–950 °C).

## 3. Results and Discussions

The glass, RHA, and calcium carbonate used in the production of the glass foams were investigated, and their potential was assessed by X-ray fluorescence (XRF) (Table 1). The results show that the raw materials have the potential to be used in the production of glass foam, in addition to being suitable for recycling. The soda-lime glass can represent up to 97% of the glass foam mass when used together with the addition of CaCO_3_—a low-cost binder that favors the volumetric expansion of glass foams [11,16,24,26,27,38]. RHA with high SiO_2_ content was incorporated into the mass of the softened glass when it presented low viscosity. The RHA contributed to the glass foam expansion, showing that it has the potential to produce this material [11]. As expected, the CaCO_3_ used in this study showed a high concentration of CaO because it is an industrialized commercial product (Table 1). The values of loss on ignition (LOI) for CaCO_3_ and RHA are shown in Figure 1, as the release of CO_2_ has potential for applications as a porogenic agent.

Figure 1 illustrates the differential thermal (DTA) and thermogravimetric (TGA) analyses of the porogenic agents. It should be noted that the RHA (Figure 1a) presented an endothermic event at approximately 100 °C, which was associated with a mass loss of 1% due to residual moisture, and an exothermic peak at 450 °C, which was associated with a 2% mass loss related to the release of volatile substances. One of these substances is phosphorus oxide, which is present in RHA, as demonstrated in the results from the chemical analysis [39,40]. Later, two other exothermic events were observed at approximately 800 and 900 °C, respectively. In the same region, it was possible to observe a mass loss of 4% associated with residual carbon decomposition into CO_2_.

Figure 1b illustrates the thermal behavior of CaCO_3_, where it can be observed that CaCO_3_ presents an exothermic event close to 700 °C, which is associated with a mass loss (40%) that is close to the expected theoretical value of 44% wt for pure CaCO_3_, resulting from the release of CO_2_ [41]. It should be considered that the appropriate firing temperature is critical for glass foam production, as it is related to glass viscosity. Glass expansion occurs through the release of gas resulting from the decomposition of the pore-forming agent when it presents adequate viscosity. The calcium carbonate reached its greatest thermal decomposition after 600 °C (Figure 1b). Glass becomes softened at higher temperatures [11]. Therefore, there is trapping of the gas in the glass matrix due to its adequate viscosity, which favors the creation of closed pores [37] and the production of glass foams [35].

Figure 2 illustrates the results of the X-ray diffraction (XRD) of the samples sintered at 850 °C, 900 °C and 950 °C. Without exception, an amorphous halo resulting from the high concentration of the glass phase can be observed. The crystalline phases were identified in all analyzed samples by the presence of cristobalite (SiO_2_) and wollastonite (CaSiO_3_). These results show the reactive condition of RHA in the soda-lime glass structure, as well as that the remaining carbon was sintered, favoring the crystallization of amorphous silica ashes in cristobalite. This result was expected, considering that the samples were sintered at temperatures that favor the formation of this silica phase, although the peak intensity was compatible with that of a system with high amorphism [42].

Figure 3 shows the volumetric expansion evolution of the glass foam during heat treatment (800–950 °C). The expansion started (800–850 °C) at the glass foam edges [43] favored by the low viscosity of soda-lime glass at these temperatures [44], facilitating the incorporation of RHA rich in SiO_2_ (89.47%) (Table 1) into the glass mass, promoting vitrification, and forming a new cellular structure [45]. The viscosity range for glass foams is usually 10^4^–10^6^ Pas [46]. The expansion was also influenced by the decomposition of CaCO_3_, generating CO_2_ at approximately 600–850 °C, as illustrated in Figure 1b [47]. From 880 °C, volumetric expansion occurred gradually with increasing temperature (950 °C). It was also possible to identify (Figure 3) the maximum thermal efficiency of CaCO_3_ that occurred at 900–950 °C [41], and from this temperature range, volumetric expansion was incipient. After analyzing the behavior of the glass foam during the heat treatment (Figure 3), the temperature interval (850–950 °C) to be investigated concerning the technological, thermal, and mechanical properties was defined. The geometric shapes of the glass foam in the green state and after sintering are illustrated in Figure 4 and Figure 5, respectively.

Figure 4 shows the geometric shape and size of the glass foam in the green state (60 × 20 × 20 mm^3^) after uniaxial pressing. Visually comparing Figure 4 and Figure 5a–c, it seems that all produced glass foams expanded significantly regardless of the temperature: 850 °C (368 ± 18%), 900 °C (453 ± 22%) and 950 °C (426 ± 21%). The volumetric expansion presented by each sintered glass foam shows that the material has the potential to be used as a coating in fire-resistant walls [48], since it was produced with smaller amounts of raw material and can cover a larger area due to its volumetric expansion potential. The glass foam expansion (Figure 5a–c) occurred because of the low viscosity of the soda-lime glass and the incorporation of RHA in its structure, increasing the reactivity of the amorphous SiO_2_ contained in these materials (Figure 2 [49] (Table 1). The incorporation of RHA into the glass foam structure was facilitated by the similarity in the particle size of the glass powder (D_90_ = 73.51 µm) and RHA (D_90_ = 73.92 µm) [50]. Reactivity between glass and ash was higher in foams from Figure 5b,c and smaller in foams from Figure 5a, though satisfactory in the latter case. Figure 5a–c show the outer surface of the glass foams and the appearance of pores that increase in size and concentration with increasing temperature, which is a result of the escape of gas (CO_2_) generated by the decomposition of CaCO_3_ into the environment. Figure 5b,c present the largest number of pores, resulting in the greatest leakage of CO_2_ due to CaCO_3_ being more efficient at these temperatures.

Figure 6 shows the results for the glass foams regarding apparent density (0.30 ± 0.009, 0.28 ± 0.009, and 0.29 ± 0.009 g/cm^3^), compressive strength (1.25 ± 0.03, 1.18 ± 0.03, and 1.21 ± 0.03 MPa), and thermal conductivity (0.326 ± 0.009, 0.279 ± 0.009, and 0.283 ± 0.009 W/mK) in relation to temperature (850, 900 and 950 °C), respectively. Figure 7 shows the results for porosity (82 ± 4, 87 ± 4, and 86 ± 4%) in relation to temperature (850, 900 and 950 °C), respectively. The results found for porosity, compressive strength, thermal conductivity, and apparent density are within the limits of those found for commercial glass foams according to the literature: porosity (85–95 ± 4%); compressive strength (0.4–6.0 MPa) [51]; thermal conductivity (0.005–0.008 W/mK) [52]; and apparent density (0.1–0.3 g/cm^3^) [53,54]. In addition to these properties, the material has the potential to be used as a wall coating, improving thermal comfort in the construction sector [48].

Figure 7 shows the architecture and pore distribution through the cross-section of each sample (Figure 7a–c) after sintering, where it is possible to observe the result of efficient CaCO_3_ decomposition at each temperature, which favored the apparent density and the new glass-ceramic structure in each foam [18]. Visual evaluation of the pores in the intersections of the walls [37] and the optical microscopy images (Figure 7a–c) evidenced predominance of well interconnected open pores, which guarantees high permeability and favors density [37], showing a strong tendency to apparent porosity because the pores provide access to the surface. This statement is supported by the type of foaming agent used, as the carbonates decompose with increasing temperature, and thus give rise to open pores. It is worth emphasizing that total porosity includes both open and closed pores. It is also observed that for each formulation, the minimum values of apparent density present depend on a balance between the decrease in viscosity and the increase in temperature, which favors the expansion of the melt material under the internal pressure of the gas (CO_2_). The release of this gas is responsible for the gradual collapse of the walls that join the pores in the foam. The low viscosity at each temperature resulted from the influence of the presence of RHA in the glass mass. Geometry of the pores at the different temperatures varied according to their location in the glass foam, presenting a spherical shape when closer to the center and the shape of rounded tetrahedral when in the corners [55]. Visually, all the pores are well interconnected and have a very homogeneous geometry, with larger pores surrounded by smaller pores without cracks [56]. There is similarity in pore morphology (Figure 7b,c), which was achieved by the influence of temperature (900–950 °C), guaranteeing maximum stability of the glass foams in the formation of their internal cellular structure [19]. The addition of RHA guaranteed heterogeneity and favored volumetric expansion and coalescence, forming a larger number of open pores, resulting in the stability and formation of the new internal cellular structure. The CO_2_ released from CaCO_3_ breaks through the walls between the pores in a similar way [57], thus increasing pore connectivity. The best results were obtained for the glass foams sintered at 900 and 950 °C: apparent density (0.28 ± 0.009 and 0.29 ± 0.009 g/cm^3^), thermal conductivity (0.279 ± 0.009 and 0.283 ± 0.009 W/mK), and porosity (87 ± 4 and 86 ± 4%, respectively). However, porosity values should be evaluated in relation to the resulting compression strength since these properties are generally inversely related [58]. This statement is supported by the best result presented in this study for compressive strength (1.25 ± 0.03 MPa), which was favored by the higher apparent density (0.30 ± 0.009 g/cm^3^) found in the glass foam sintered at 850 °C (Figure 6). These values are derived from the foam morphology (Figure 7a), which presents a smaller pore size (82 ± 4 %). Figure 6 with pores very close to each other, increasing the degree of adhesion between the soda-lime glass, RHA, and CaCO_3_ interfaces, conditions that did not occur at the other temperatures.

## 4. Conclusions

This study has successfully identified the potential of soda-lime glass powder residue and the favorable influence of rice husk ash in the mixture to produce sintered glass foams at temperatures of 850, 900 and 950 °C with technological properties of commercial foams such as apparent density (0.28–0.30 g/cm^3^), porosity (82–87 ± 4%), compressive strength (1.18 ± 0.03–1.25 ± 0.03 MPa), and thermal conductivity (0.283–0.326 W/mK). The foam produced can be used to coat walls and fire doors in civil construction because it has good thermal insulation, reducing heat conduction between areas. The glass foams produced in this study are promising, as they are more economical and efficient to manufacture, resulting in lightweight materials with thermal insulation properties. In addition, they can help mitigate the problems arising from the disposal of glass waste and rice husk ash in the environment, reduce the use of natural and synthetic raw materials, enable consecutive circulation, and add value to these wastes by transforming the glass and rice husk ash into co-products, favoring the region’s circular economy.

## Figures and Tables

**Figure 1 materials-15-06669-f001:**
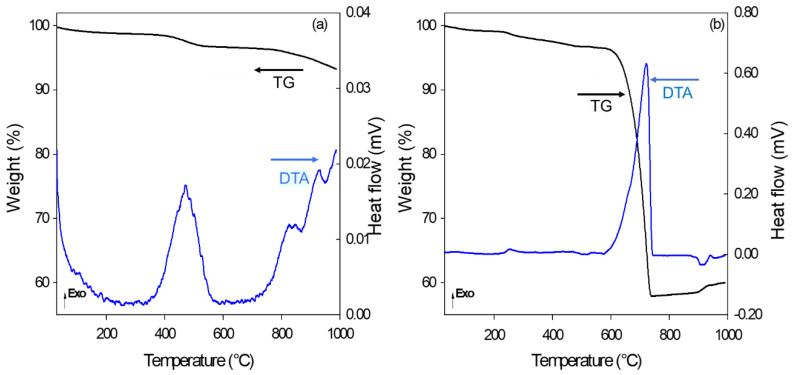
Differential thermal (DTA) and thermogravimetric (TGA) analyses of RHA (**a**) and CaCO_3_ (**b**).

**Figure 2 materials-15-06669-f002:**
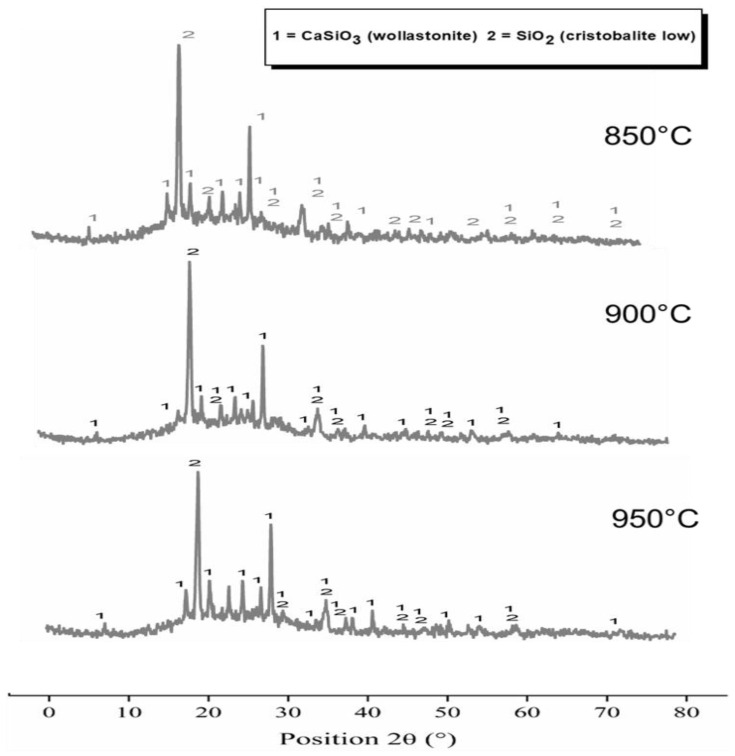
XRD patterns of samples sintered at 850, 900 and 950 °C.

**Figure 3 materials-15-06669-f003:**
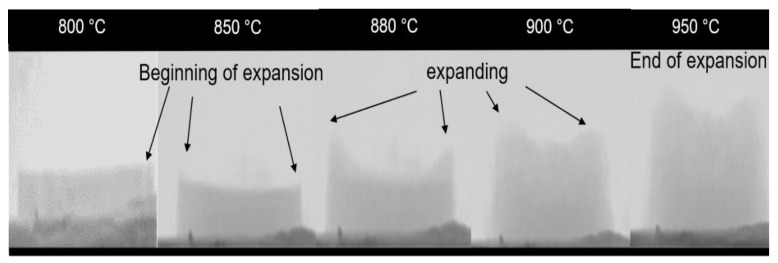
Evolution of the volumetric expansion within the temperature interval of 800–950 °C.

**Figure 4 materials-15-06669-f004:**
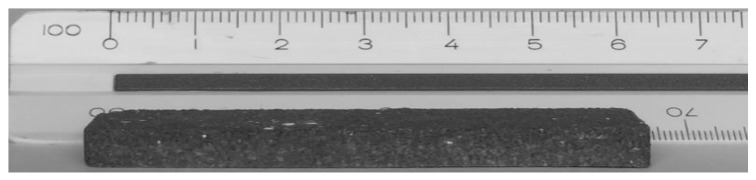
Geometric formation of glass foam in the green state.

**Figure 5 materials-15-06669-f005:**
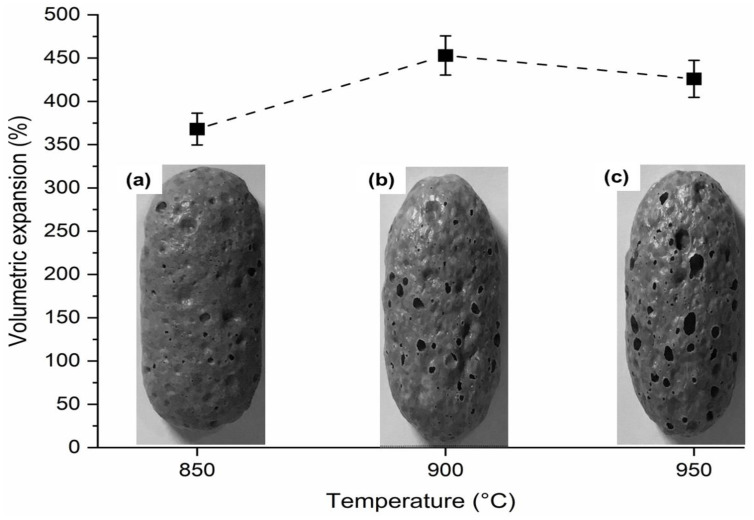
Geometric shape and volumetric expansion of the glass foams sintered at (**a**) 850 °C, (**b**) 900 °C and (**c**) 950 °C.

**Figure 6 materials-15-06669-f006:**
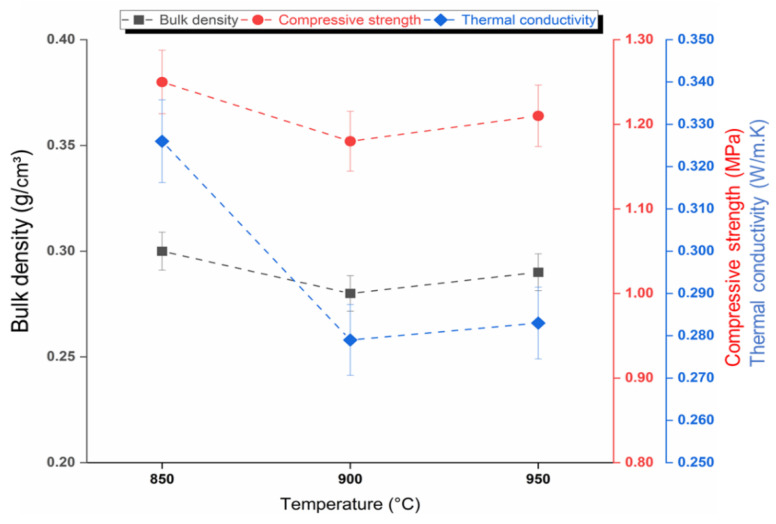
Bulk density, compressive strength, and thermal conductivity of the sintered foams: 850, 900 and 950 °C.

**Figure 7 materials-15-06669-f007:**
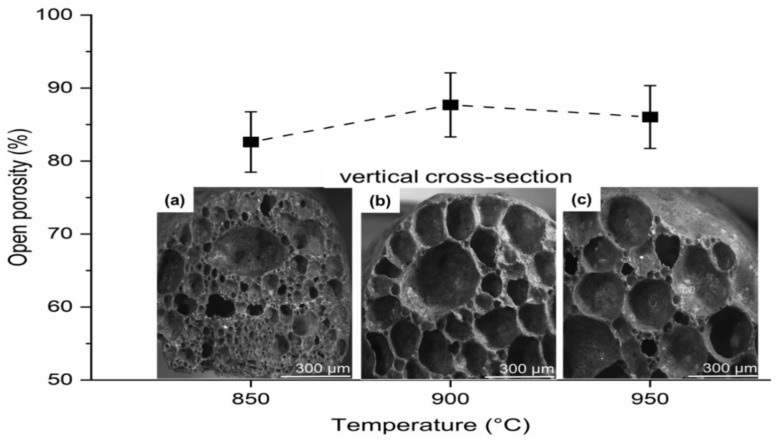
Porosity and geometric formation of pores of glass foams sintered at (**a**) 850 °C, (**b**) 900 °C, and (**c**) 950 °C.

**Table 1 materials-15-06669-t001:** Chemical analyses of glass, RHA, and CaCO_3_.

Material	Composition *						
	SiO_2_	CaO	Na_2_O	Al_2_O_3_	K_2_O	Fe_2_O_3_	P_2_O_5_
Glass	72.23	21.11	12.62	1.47	0.89	0.78	–
RHA	89.47	2.68	1.61	0.97	2.68	0.33	0.97
CaCO_3_	0.41	97.79	–	0.07	–	0.15	1.07

* Expressed in oxides. MnO, MgO, SrO, and SO_3_ found at smaller proportions.

## Data Availability

Data available on request.

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
