# Peer review of "Influence of Sintering on Thermal, Mechanical and Technological Properties of Glass Foams Produced from Agro-Industrial Residues"

_materials, 2022, doi:10.3390/ma15196669_

Round 1

Reviewer 1 Report (Previous Reviewer 3)

MAIN IMPRESSIONS

This paper has an undeniable practical usefulness. However, from a scientific point of view, the following issues must be addressed: i) Reviewed findings should be discussed in deep, ii) the novelty of the paper should be underlined and iii) It is recommended to underscore the novelty of this paper in the conclusion.

MORE DETAILED COMMENTS

Line 3: superscript in cm3.

Line 18: subscript in SiO2, Na2O, SiO2, Na2O, Al2O3, K2O and Fe2O3.

Lines 105-106: subscripts.

Line 104: Table 1 does not follow the rules established by mdpi.

Line 104: Table 1: Could you please add the Na2O content?

Lines 339-350: Delete:

1. Author 1, A.B.; Author 2, C.D. Title of the article. Abbreviated Journal Name Year, Volume, page range. 339

2. Author 1, A.; Author 2, B. Title of the chapter. In Book Title, 2nd ed.; Editor 1, A., Editor 2, B., Eds.; Publisher: Publisher Location, 340 Country, 2007; Volume 3, pp. 154–196. 341

3. Author 1, A.; Author 2, B. Book Title, 3rd ed.; Publisher: Publisher Location, Country, 2008; pp. 154–196. 342

4. Author 1, A.B.; Author 2, C. Title of Unpublished Work. Abbreviated Journal Name year, phrase indicating stage of publication (sub-343 mitted; accepted; in press). 344

5. Author 1, A.B. (University, City, State, Country); Author 2, C. (Institute, City, State, Country). Personal communication, 2012. 345

6. Author 1, A.B.; Author 2, C.D.; Author 3, E.F. Title of Presentation. In Proceedings of the Name of the Conference, Location of 346 Conference, Country, Date of Conference (Day Month Year). 347

7. Author 1, A.B. Title of Thesis. Level of Thesis, Degree-Granting University, Location of University, Date of Completion. 348

8. Title of Site. Available online: URL (accessed on Day Month Year).

Line 354: In “Materials (Basel). 2022, 15, 5232, 354 “, delete “(Basel).”.

Line 363: subscript in CO2.

FINAL REMARK

Could you please be sure to proofread it carefully before sending it in again?

RECOMMENDATION

In conclusion, Major changes have been proposed.

Author Response

We would like to thank all the reviewers for all the support and guidance provided.  The authors made a great effort to meet the expectations and suggestions, although this article was produced at a very difficult time in Brazil and around the world, during the covid 19 pandemic, when all the laboratories were closed.  Even in this scenario, we could obtain some results to be published in this important journal.

Reviewer 2 Report (Previous Reviewer 1)

Manuscript may please be accepted.

Author Response

We would like to thank all the reviewers for all the support and guidance provided.  The authors made a great effort to meet the expectations and suggestions, although this article was produced at a very difficult time in Brazil and around the world, during the covid 19 pandemic, when all the laboratories were closed.  Even in this scenario, we could obtain some results to be published in this important journal.

Reviewer 3 Report (New Reviewer)

The manuscript describes an investigation of the technological, thermal and mechanical properties of glass  foams produced with soda-lime glass residues and rice husk ash sintered at 850-950 °C.

The topic is not novel as this has been done before; but to find the optimal sintering temperature is informative for future development; the methods are appropriate and the scope is adequate.

The manuscript is well prepared and is suitable for publication after minor revision to address some minor issues.

The title, maybe better specify from which residues.

Word order in title “thermal, mechanical and technological” is different from in abstract.

L56, “However” is not the right word.

L 93, how was the mixture “homogenized”?

L99,  “30-min threshold”?

L102, “After the compressive strength tests,” is not correct.

Table 1, proper chemical formula format with underscripted number should be used. **does not appear in Table.

Section 2.2 and 2.3 are exactly the same.

Explanation of equation parameters, “where (ρa ) is apparent density and (ρt =2.5g/cm3) is the theoretical density of the powder.” Should appear beneath the equation.

How was geometric dimension determined should be introduced briefly.

L174 “emonstrated”?

L196, “an amorphous halo” please explain where in Fig. 2 does one observe this.

Fig. 5 may be more informative to mark length and width at widest points

Fig. 6 better move “Compressive strength” right next to its axis.

L260-267 this statement has been stated in Conclusion. Remove here.

L290 plural form should be tetrahedral.

Ref 1-8 are not proper refs, delete.

Author Response

We would like to thank all the reviewers for all the support and guidance provided.  The authors made a great effort to meet the expectations and suggestions, although this article was produced at a very difficult time in Brazil and around the world, during the covid 19 pandemic, when all the laboratories were closed.  Even in this scenario, we could obtain some results to be published in this important journal.

Round 2

Reviewer 1 Report (Previous Reviewer 3)

Line 107: subscript in Al2O3 (Table 1).

Could you please change ºC by °C throughout the text? For instance: Line 165: In 700ºC should be °C, and Line 171: In 600ºC should be °C.

This manuscript is a resubmission of an earlier submission. The following is a list of the peer review reports and author responses from that submission.

Round 1

Reviewer 1 Report

The authors investigated the foam glasses formed using glass powder, CaCO3 and rice husk ash. Preliminary characterizations were done for the source materials. Foam glasses were formed through various optimizations and compositions of the precursors. The following are the major suggestions that the authors may consider. The key research hypothesis of this work should be included in the abstract. The purpose of this work (and the novelty) shall be included in the abstract and in the last paragraph of the introduction. In fact, the last paragraph of the introduction shall be improved by highlighting the key results and important conclusions. Carbonization extent in RHA should be studied and included in the revised manuscript. Fig 4 should be explained clearly. The insets in figure 4 correspond to SEM image? This requires further explanation and importantly, scale bar. Physical significance and quality of discussions arrived from figures 2,3 and 4 should be improved. These changes shall be incorporated and highlighted in the revised submission.

Reviewer 2 Report

Dear authors interesting work, however you need to clearly explain your findings with reference to literature. Need to explain how you got superior mechanical and physical properties with the addition of agro-industrial waste into foam glasses to achieve better thermal insulations.

Need to check your formatting as well. Some sentences are too long and introduction's first paragraph is too long for the reader. 

Need to add some literature as suggested and convinced readers of your research outputs by presenting more discussions in the paper. 

Also please add the significance of the current research and literature gap, which is currently missing.

Please check pdf attached for my detailed comments. 

Reviewer 3 Report

Ms. Ref. No.: materials-1698202-peer-review-v1

Foam glass with the addition of agro-industrial waste for use as thermal insulation material

Reviewer comments:

SUMMARY

Rice husk is usually used for wall and roof insulation, and it has been used as insulating material for cold storage for many years in China, because is available locally. The manuscript is a good paper assessing the foam glass production by adding agro-industrial wastes for use as thermal insulation applications. This is a topic that has not been widely covered in the literature, therefore, this a subject of great interest, but it is somehow limited in the analysis and application of these results.

MAIN IMPRESSIONS

This paper has an undeniable practical usefulness. However, from a scientific point of view, the following issues must be addressed: i) Reviewed findings should be discussed in deep, ii) the novelty of the paper should be underlined and iii) It is recommended to underscore the novelty of this paper in the conclusion.

MORE DETAILED COMMENTS

Editorial: This paper does not follow the rules established by mdpi. Could you please use the materials-mdpi template? It is quite difficult to review the paper without numbering the pages:

https://www.mdpi.com/journal/materials/instructions#preparation

Abstract Lines … :  Subscript is missing in “ …SiO2, Na2O  … in SiO2 (CaO, Na2O, Al2O3, K2O, Fe2O3) …”.

  1. Introduction - Lines 9, 12 and so on: “foam glass materials” could be better than “foam glasses”.

Page 8 of 19: Obviously  ”...the CaCO3 showed a high concentration of CaO …”, but also CO2. Could you please add the CO2 content in Table 1? This calcium carbonate does not present any magnesium? This calcium carbonate is calcite, vaterite, aragonite, limestone, …?

Page 11 of 19: Could you please provide the compressive strength (Kg/cm²) as MPa or N/mm2?

Page 13 of 19: It is recommended to underscore the novelty of this paper in the conclusion.

Page 13 of 19: Subscript is missing in “…addition of CaCO3 …”.

  • Page 13 of 19: 3. Conclusion: Could you please add the residual rice husk ash content in the mix design? It is recommended to underscore the novelty of this paper in the conclusion. Add the optimum mix design. Could you please rewrite the conclusions?

References

References must be listed individually at the end of the manuscript following this format:

  1. Author 1, A.B.; Author 2, C.D. Title of the article. Abbreviated Journal Name Year, Volume, page range.
  2. Author 1, A.; Author 2, B. Title of the chapter. In Book Title, 2nd ed.; Editor 1, A., Editor 2, B., Eds.; Publisher: Publisher Location, Country, 2007; Volume 3, pp. 154–196.

RECOMMENDATION

In conclusion, Major changes have been proposed.